# Impacts of Postoperative Adjuvant Therapies on the Survival of Women with High-Risk Early-Stage Endometrial Cancer: A Cohort Study

**DOI:** 10.3390/cancers17020187

**Published:** 2025-01-08

**Authors:** Hee Joong Lee, Banghyun Lee

**Affiliations:** 1Department of Obstetrics & Gynecology, Uijeongbu St. Mary’s Hospital, College of Medicine, The Catholic University of Korea, Seoul 11765, Republic of Korea; heejoong@catholic.ac.kr; 2Department of Obstetrics and Gynecology, Inha University Hospital, College of Medicine, Inha University, Incheon 22332, Republic of Korea

**Keywords:** endometrial cancer, adjuvant therapy, radiotherapy, chemotherapy, hormone therapy

## Abstract

The survival outcomes in women with high-risk early-stage endometrial cancer (EEC) depend on various postoperative adjuvant therapeutic strategies. In this Korean nationwide cohort study, the impacts are evaluated of various postoperative adjuvant therapies on the survival of women with high-risk EEC. Except for a significant small effect of adjuvant radiotherapy after chemotherapy, postoperative adjuvant therapies had similar impacts on the overall survival of women with high-risk EEC. These results provide helpful information for selecting postoperative adjuvant therapy in women with high-risk EEC.

## 1. Introduction

Globally, endometrial cancer is the most prevalent genital tract cancer in women [1]. In South Korea, it has shown an annual increase of 4.7% between 1999 and 2019, with 6.8% of cases in women <30 years of age [2].

The standard treatment of endometrial cancer is staging surgery and various adjuvant therapies can be performed [3]. Radiotherapy with a locoregional effect may prevent and manage local recurrence and chemotherapy with a systemic effect may prevent and manage distant metastasis [4,5]. High-risk early-stage endometrial cancer (EEC) for recurrence is defined as International Federation of Gynaecology and Obstetrics (FIGO) 2009 stage I, grade 3 endometrioid histology with myometrial invasion >50% or lymphovascular space invasion (LVSI) or both; stage I, non-endometrioid histology; or stage II [6,7,8,9,10,11]. Women with high-risk EEC are candidates for postoperative adjuvant therapies [6,7,8,9,10,11].

Recently, two randomized controlled trials (RCTs) reported the impacts of postoperative adjuvant therapy on the survival of women with high-risk EEC [8,9]. In the PORTEC-3 trial, the five-year failure-free survival (FFS) for women with high-risk stage I–II was similar in those given chemoradiotherapy (external beam radiation therapy (EBRT) ± vaginal brachytherapy (VB) plus concurrent and adjuvant chemotherapy) and EBRT ± VB [8]. In GOG 249, the five-year recurrence-free survival (RFS) and overall survival (OS) for women with high-risk stage I–II were similar in those given VB plus chemotherapy and EBRT ± VB [9]. In addition, a large-scale cohort study (n = 11,746) showed that the OS for women with high-risk EEC was improved in the women who had received adjuvant chemoradiotherapy (concurrent or adjuvant chemotherapy) compared to adjuvant radiotherapy (EBRT, VB, or both) [12].

Currently, the guidelines recommend that women with high-risk EEC can be treated using postoperative adjuvant therapy such as radiotherapy (EBRT, VB, or both) ± (concurrent or adjuvant chemotherapy or both), chemotherapy ± EBRT ± VB, or EBRT ± VB [7,13]. On the other hand, the survival outcomes among those postoperative adjuvant therapeutic strategies have not been clearly compared, suggesting the necessity of further evaluation. Therefore, the impacts of various postoperative adjuvant therapies on the survival of women with high-risk EEC were investigated in a Korean nationwide cohort study.

## 2. Materials and Methods

In South Korea, the National Health Service, a universal health coverage system, covers 98% of the population [14,15]. The Korean Health Insurance Review and Assessment Service (HIRA) shares most of the National Health Insurance Service data. Recently, the claims data of the HIRA were connected with Korean cancer registration data. Using these data, in this study, the data were analyzed from women with endometrial cancer diagnostic codes registered with the HIRA and diagnosed with endometrial cancer based on pathology between 1 January 2007 and 31 December 2019. This study was approved by the Institutional Review Board of Inha University Hospital (No. 2022-05-036) on 19 December 2022. The HIRA dataset and Korean cancer registration data use anonymous identifying numbers to safeguard personal data, in accordance with South Korea’s Bioethics and Safety Act. Therefore, it was not necessary to obtain informed consent from the patients who were part of the data.

For subject selection and analysis, in this study, the Korea Health Insurance Medical Care Expenses (2017 and 2019 versions), the International Classification of Diseases, the 10th revision (ICD-10), and the HIRA Drug Ingredients Codes were used. The definition of endometrial carcinoma in women was women diagnosed with endometrial cancer based on the pathology from primary surgery between 2007 and 2019. Those patients had the diagnostic codes for endometrial cancer (ICD-10: C54x or C55). Women with early-stage endometrial cancer were defined as women diagnosed with endometrial cancer confined to the uterus based on the pathology from staging surgery, including primary surgery. Among women with early-stage endometrial cancer, women with the diagnostic codes for other cancers within two years before primary surgery were excluded. The following exclusion criteria were applied: women diagnosed with venous thromboembolism (VTE) before primary surgery for endometrial cancer and women ineligible for high-risk EEC (Figure 1).

Women with adjuvant EBRT ± VB, adjuvant chemotherapy, adjuvant chemoradiotherapy, adjuvant hormone therapy, or non-endometrioid histologic types were considered to have high-risk EEC.

Regarding medical protection, the term “low socioeconomic status” (SES) was used to describe the type of insurance. The Charlson Comorbidity Index (CCI) was calculated using the data obtained between 365 days and 1 day before the first diagnostic date of endometrial cancer [16]. The histological types consisted of endometrioid adenocarcinoma and non-endometrioid adenocarcinoma, including serous adenocarcinoma, clear cell adenocarcinoma, undifferentiated/dedifferentiated carcinoma, and carcinosarcoma. Primary surgery was defined using the surgery codes for a total hysterectomy or radical hysterectomy and the presence of a simultaneous pathology report for endometrial cancer. Adjuvant therapy was defined as radiotherapy, chemotherapy, chemoradiotherapy, or hormone therapy within 12 weeks after the primary surgery. Radiotherapy was defined as the prescription codes for EBRT or VB with a simultaneous diagnostic code for endometrial cancer. Chemotherapy was defined as the prescription codes for chemotherapy (cisplatin, carboplatin, dacarbazine, docetaxel, doxorubicin, epirubicin, everolimus, gemcitabine, ifosfamide, liposomal doxorubicin, paclitaxel, pazopanib, trabectedin, temozolomide, temsirolimus, and topotecan) with a simultaneous diagnostic code for endometrial cancer. Chemoradiotherapy consisted of concurrent chemoradiotherapy (CCRT) and sequential therapy, including chemotherapy after radiotherapy and radiotherapy after chemotherapy with a simultaneous diagnostic code for endometrial cancer. CCRT was defined as the codes for EBRT or VB and the simultaneous codes for chemotherapy. Hormone therapy was defined using prescription codes such as progestins, tamoxifen, progestins and tamoxifen, aromatase inhibitors, and gonadotropin-releasing hormone agonists with the simultaneous diagnostic code for endometrial cancer. Women with VTE were defined as women who received prescriptions for anticoagulants more than twice simultaneously with the diagnostic codes for VTE (deep vein thrombosis, pulmonary embolism, or both) after primary surgery. Prescription codes for anticoagulants more than twice without the VTE diagnostic codes after primary surgery were considered prophylactic anticoagulant use for VTE. Unfractionated heparin, low molecular weight heparin, warfarin, and direct oral anticoagulants were the anticoagulants used (Appendix A).

### Statistical Analyses

Big data were explored, modified, and analyzed using SAS version 9.4 (SAS Institute Inc., Cary, NC, USA). The relationships between variables and OS were examined using univariate and multivariate analyses through the Cox proportional hazards regression model. The cumulative probability of OS was examined by building the Kaplan–Meier curves using the log-rank test. Two-tailed tests were used for all statistical analyses, and *p* values < 0.05 were considered significant.

## 3. Results

Initially, the data of 10,901 women diagnosed with early-stage endometrial cancer from the Korean cancer registration data and registered by the HIRA between 2007 and 2019 were extracted. The data of 1341 women who met the eligibility criteria for high-risk EEC were selected from this cohort (Figure 1).

### 3.1. Baseline Characteristics

Table 1 lists the baseline characteristics of the study subjects.

The study subjects were followed up for 5.9 ± 4.0 years. Among the 26 women who received adjuvant VB alone, nineteen, two, and five women had serous adenocarcinoma, undifferentiated/dedifferentiated carcinoma, and carcinosarcoma. Appendix A lists the characteristics of 18 women who received adjuvant chemoradiotherapy. Appendix A list the distribution of treatment types of adjuvant radiotherapy and chemoradiotherapy in women with endometrioid or non-endometrioid histology, respectively. Women who received primary surgery alone, adjuvant VB alone, adjuvant CCRT, or adjuvant radiotherapy after chemotherapy consisted only of those with a non-endometrioid histology (Appendix A).

### 3.2. Associations Between Types of Treatments and OS

After adjusting for the confounding variables, the OS was similar in the women who had received adjuvant EBRT ± VB, adjuvant VB alone, adjuvant chemotherapy, or adjuvant hormone therapy in combination with primary surgery compared to those who had received primary surgery alone. On the other hand, the OS was significantly lower in those given a combination of primary surgery and adjuvant chemoradiotherapy than in those given primary surgery alone (HR 3.083; 95% CI 1.311–7.247; *p* = 0.010). Moreover, the OS decreased significantly with age (HR 1.051; 95% CI 1.033–1.069; *p* < 0.001). The postoperative VTE was significantly associated with a lower OS (HR 2.75; 95% CI 1.675–4.513; *p* < 0.001). Prophylactic anticoagulants were not associated with the OS (Table 2).

In women with endometrioid histology (n = 794), all women received adjuvant therapies (adjuvant EBRT ± VB, adjuvant chemotherapy, adjuvant chemoradiotherapy, or adjuvant hormone therapy) except for adjuvant VB alone and no women received primary surgery alone (Appendix A). When analyzed in women with endometrioid type, the OS was similar according to these types of treatments (Figure 2A). Over the follow-up periods of women with an endometrioid histology, 85.2% (397/466), 95.3% (221/232), 83.3% (5/6), and 95.6% (86/90) of women who received adjuvant EBRT ± VB, adjuvant chemotherapy, adjuvant chemoradiotherapy, and adjuvant hormone therapy, respectively, were alive. On the other hand, analysis of women with non-endometrioid histology showed that the OS differed according to the types of treatments (*p* = 0.021) (Figure 2B). Over the follow-up periods of women with non-endometrioid histology (n = 547), 80.9% (242/299), 57.1% (28/49), 84.6% (22/26), 86.3% (138/160), 58.3% (7/12), and 100% (1/1) of the women who received primary surgery alone, adjuvant EBRT ± VB, adjuvant VB alone, adjuvant chemotherapy, adjuvant chemoradiotherapy, and adjuvant hormone therapy, respectively, were alive.

### 3.3. Associations Between Adjuvant Radiotherapy and Chemoradiotherapy and OS

After adjusting for the confounding variables, the OS was significantly lower in the adjuvant radiotherapy after chemotherapy group (HR 11.87; 95% CI 4.595–30.664; *p* < 0.001) than in the adjuvant EBRT ± VB group. On the other hand, the OS was similar in the adjuvant VB alone, adjuvant CCRT, or adjuvant chemotherapy after radiotherapy groups to the adjuvant EBRT ± VB group. Moreover, the OS decreased significantly with age (HR 1.068; 95% CI 1.047–1.099; *p* < 0.001). The postoperative VTE was associated with a decreased OS (HR 3.441; 95% CI 1.65–7.176; *p* = 0.001). Prophylactic anticoagulants were not associated with the OS (Table 3). In addition, other chemotherapeutic agents were significantly associated with a decreased OS compared to the platinum-based agents (HR 3.398; 95% CI 1.764–6.545; *p* = 0.001).

Women with the endometrioid histological type received only adjuvant EBRT ± VB and adjuvant chemotherapy after radiotherapy (Appendix A). Those therapies produced a similar OS (Figure 3A). Over the follow-up periods of women with endometrioid histology, 85.2% (397/466) and 83.3% (5/6) of the women who received adjuvant EBRT ± VB and adjuvant chemotherapy after radiotherapy, respectively, were alive. On the other hand, an analysis of women with non-endometrioid histology showed a different OS according to the types of adjuvant radiotherapy and adjuvant chemoradiotherapy (*p* = 0.023) (Figure 3B). Over the follow-up periods of women with a non-endometrioid histology, 57.1% (28/49), 84.6% (22/26), 100% (2/2), 100% (2/2), and 37.5% (3/8) of the women who received adjuvant EBRT ± VB, adjuvant VB alone, adjuvant CCRT, adjuvant chemotherapy after radiotherapy, and adjuvant radiotherapy after chemotherapy, respectively, were alive.

## 4. Discussion

Multivariate analysis adjusted for the co-variables showed that in women with high-risk EEC, the OS was similar in various adjuvant therapies to those given primary surgery alone, except for a significant decrease in adjuvant chemoradiotherapy. Moreover, compared with adjuvant EBRT ± VB, the OS was significantly lower in the women who had received adjuvant radiotherapy after the chemotherapy but was similar in those who had received adjuvant VB alone, adjuvant CCRT, and adjuvant chemotherapy after radiotherapy. Furthermore, in women with endometrioid type, the OS was similar regardless of the types of treatments and types of adjuvant radiotherapy and chemoradiotherapy. In contrast, in women with non-endometrioid types, the OS differed according to the types of treatments and types of adjuvant radiotherapy and chemoradiotherapy.

Although it is classified into EEC (FIGO 1988 or 2009 stage I–II), high-risk EEC is associated with an increased risk of vaginal or pelvic recurrence and distant metastases. In women with high-risk EEC, the effect of postoperative adjuvant therapy, such as radiotherapy, chemotherapy, or chemoradiotherapy, is debatable.

Three RCTs (PORTEC-1, GOG-99, and ASTEC/EN5) reported that in women with high-intermediate risk or high-risk EEC, adjuvant EBRT compared with no adjuvant therapy decreased the locoregional recurrence, but without an improvement in OS [17,18,19]. A large-scale cohort study (n = 1709) showed that the OS for women with uterine serous carcinoma confined to the endometrium was similar in those who had received adjuvant radiotherapy (EBRT, VB, or both) compared to the women who had received no adjuvant therapy, but it was improved in those who had received adjuvant chemotherapy and adjuvant chemoradiotherapy [20]. A retrospective study (n = 112) showed that the RFS and OS for women with early-stage uterine clear cell carcinoma were not improved by any adjuvant therapy (pelvic radiation or CCRT, VB, chemotherapy, and chemoradiotherapy with pelvic radiation or VB) compared to the no adjuvant therapy group [21]. In the present study, women who received primary surgery alone without adjuvant therapy and those who received adjuvant VB alone consisted only of women with a non-endometrioid histology (Appendix A). The OS for women with high-risk EEC was similar in those given adjuvant EBRT ± VB and adjuvant VB alone, compared to those given no adjuvant therapy, corresponding to the results of previous studies [17,18,19,20,21]. In the present study, however, compared with the women who had received no adjuvant therapy, adjuvant chemotherapy produced a similar OS, and adjuvant chemoradiotherapy showed a shorter OS. In addition, when analyzed in women according to the endometrioid type, the survival rates were similar in the women who had received adjuvant EBRT ± VB, adjuvant chemotherapy, adjuvant chemoradiotherapy, and adjuvant hormone therapy. On the other hand, in women with the non-endometrioid histology type, the survival rates of adjuvant EBRT ± VB and adjuvant chemoradiotherapy were lower than in those given primary surgery alone, adjuvant VB alone, adjuvant chemotherapy, and adjuvant hormone therapy. These findings were different from the results of previous studies, suggesting the necessity of further studies [20,21]. In addition, in this paper, a similar OS is reported in those given adjuvant hormone therapy (endometrioid, n = 90; non-endometrioid, n = 1) or no adjuvant therapy. This result suggests the necessity of studies to investigate the impact of adjuvant hormone therapy in women with high-risk EEC (Table 1, Table 2, and Appendix A).

Two RCTs (Norwegian trial and PORTEC-2) reported that in women with intermediate-risk or high-intermediate-risk EEC, adjuvant VB alone had a similar OS to those given adjuvant EBRT ± VB [22,23]. A large-scale cohort study showed that in women with high-risk EEC, the OS for adjuvant VB alone (n = 3892) was similar to those given adjuvant EBRT (n = 2338) [12]. In a retrospective study (n = 122), the OS for women with high-risk EEC was similar to those given adjuvant VB alone and adjuvant EBRT ± VB, but there were benefits to the RFS and disease-free survival (DFS) in EBRT ± VB [24]. In the present study, the OS for women with high-risk EEC was similar in adjuvant VB alone (only non-endometrioid, n = 26) compared to adjuvant EBRT ± VB, corresponding to the results of previous studies [12,22,23]. Nevertheless, when analyzed in women with the non-endometrioid histology type, the survival rate of adjuvant VB alone was higher than in those given adjuvant EBRT ± VB. The extremely low rate of women who received adjuvant VB alone can be attributed to these findings (Table 1, Table 3, and Appendix A).

Two RCTs reported that in women with high-risk EEC, those given adjuvant chemotherapy after radiotherapy ± CCRT had similar survival rates to those given adjuvant EBRT ± VB [8,9]. In the PORTEC-3 trial, the FFS was similar in the EBRT ± VB plus concurrent and adjuvant chemotherapy group than those in the adjuvant EBRT ± VB group [8]. In GOG 249, the RFS and OS were not different in the VB plus adjuvant chemotherapy compared with adjuvant EBRT ± VB [9]. A large-scale cohort study showed that in women with high-risk EEC, adjuvant chemoradiotherapy (CCRT and chemotherapy after radiotherapy) (n = 3540) was associated with an improved OS compared to those given adjuvant radiotherapy (EBRT, VB, or both) (n = 8206), but the OS was similar in those given CCRT (67.2%) and chemotherapy after radiotherapy (32.8%) [12]. A retrospective study (n = 64) showed that the RFS and OS for women with EEC were similar in those given adjuvant radiotherapy (EBRT + VB) and adjuvant CCRT (EBRT + VB plus weekly cisplatin) [25]. In another retrospective study, the progression-free survival and OS for women with high-risk EEC (endometrioid type, n = 116) were improved by adjuvant chemoradiotherapy compared to adjuvant radiotherapy (EBRT, VB, or both) [26]. In the present study, the OS for women with high-risk EEC was similar in those given adjuvant CCRT and adjuvant chemotherapy after radiotherapy compared to those given adjuvant EBRT ± VB. In addition, when the women with endometrioid type were analyzed, the survival rates were similar in adjuvant EBRT ± VB and adjuvant chemotherapy after radiotherapy. On the other hand, in women with non-endometrioid histology type, survival rates of adjuvant VB alone, adjuvant CCRT, and adjuvant chemotherapy after radiotherapy were higher than in those given adjuvant EBRT ± VB. These findings can be attributed to the extremely low number of women who received chemoradiotherapy (Table 1, Table 3, and Appendix A).

A retrospective study (n = 414) showed that disease-specific survival (DSS) and OS for women with high-risk EEC were not improved by adjuvant radiotherapy (EBRT, VB, or chemoradiotherapy) or adjuvant chemotherapy ± radiotherapy. In contrast, the RFS was not improved by adjuvant radiotherapy, but it was improved by adjuvant chemotherapy [22]. In addition, the RFS, DSS, and OS for women with high-risk EEC (endometrioid type, n = 60) were not improved by both adjuvant therapies [27]. In another retrospective study, the DFS and OS for women with high-risk EEC (endometrioid type, n = 145) were similar between adjuvant radiotherapy (EBRT, VB, or both) ± chemotherapy and adjuvant chemotherapy [11]. In the present study, the OS was significantly lower in those who received adjuvant radiotherapy after chemotherapy (only non-endometrioid, n = 8) than in those who received adjuvant EBRT ± VB. In addition, when analyzed in women with the non-endometrioid histology type, the survival rate of adjuvant radiotherapy after chemotherapy was lower than in those who received adjuvant EBRT ± VB. Women who receive adjuvant radiotherapy after chemotherapy might have an increased risk of local aggravation during chemotherapy. Nevertheless, these findings suggest that the extremely low rates of women who received adjuvant radiotherapy after chemotherapy can be responsible for low OS (Table 1, Table 3, and Appendix A).

In the present study, the OS decreased significantly with age, corresponding to a large-scale cohort study (n = 1764) in which age was a significant predictor of recurrence, death from disease, and death in women with endometrial cancer [28]. The VTE has been reported to be one of the leading causes of death in women with cancer [29]. In the present study, although the incidence of VTE was very low (3.8%), postoperative VTE was significantly associated with a decreased OS, showing that VTE is a poor prognostic factor for cancer. On the other hand, in the present study, prophylactic anticoagulants were not associated with the OS, highlighting the need for further studies (Table 2 and Table 3). In addition, the present study showed that other chemotherapeutic agents were significantly associated with a lower OS compared to platinum-based agents, supporting the guideline that platinum-based agents are the preferred agents in women with endometrial cancer [7,13].

This population-based cohort study, conducted nationwide, was the first to investigate how different postoperative adjuvant therapies affected the survival of women with high-risk EEC. This study had the following limitations. First, diagnostic and prescription codes were used to characterize the diseases (except for endometrial cancer) and treatments without looking through the medical records. Therefore, some women with the wrong codes might have been misidentified. Second, the data did not include information on the grade and stage because the HIRA claim dataset related to cancer registration data does not provide information on the grade, stage, and surgical findings. Nevertheless, it was assumed that most of the women with endometrioid histology might be “stage I patients with grade 3 endometrioid histology, associated with myometrial invasion ≥50% or LVSI, and stage II patients” because all women with an endometrioid histology received adjuvant EBRT ± VB, adjuvant chemotherapy, adjuvant chemoradiotherapy, or adjuvant hormone therapy after primary surgery (Figure 2). Finally, there were extremely low rates of women with high-risk EEC who received adjuvant VB alone or adjuvant chemoradiotherapy. These may limit the significance of the results.

## 5. Conclusions

An analysis of the claim data related to cancer registration data showed that except for adjuvant radiotherapy after chemotherapy, postoperative adjuvant therapies have similar impacts on the OS of women with high-risk EEC. Moreover, in women with the endometrioid type, the OS was similar in various adjuvant therapies. In contrast, the OS differed according to various adjuvant therapies in women with non-endometrioid types. Although the rates of women who received adjuvant VB alone or adjuvant chemoradiotherapy were extremely low, these results may provide helpful information for selecting postoperative adjuvant therapy in women with high-risk EEC. Nevertheless, large-scale studies will be needed to validate these findings.

## Figures and Tables

**Figure 1 cancers-17-00187-f001:**
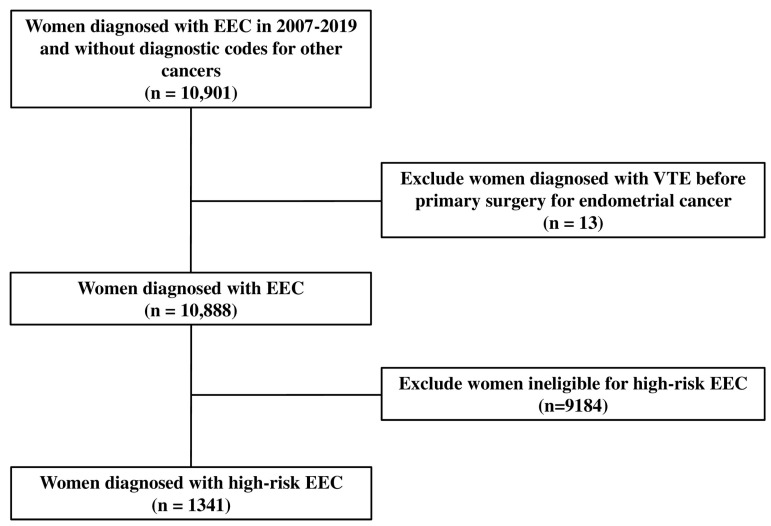
Flow chart for extracting the eligible study subjects. EEC, early-stage endometrial cancer; VTE, venous thromboembolism.

**Figure 2 cancers-17-00187-f002:**
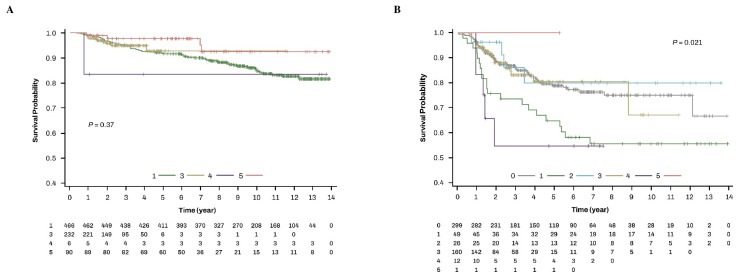
OS of types of treatments according to the type of histology in women with high-risk EEC. (**A**) endometrioid type and (**B**) non-endometrioid types. 0. Primary surgery alone; 1. primary surgery + adjuvant EBRT ± VB; 2. primary surgery + adjuvant VB alone; 3. primary surgery + adjuvant chemotherapy; 4. primary surgery + adjuvant chemoradiotherapy; 5. primary surgery + adjuvant hormone therapy. EBRT, external beam radiation therapy; EEC, early-stage endometrial cancer.

**Figure 3 cancers-17-00187-f003:**
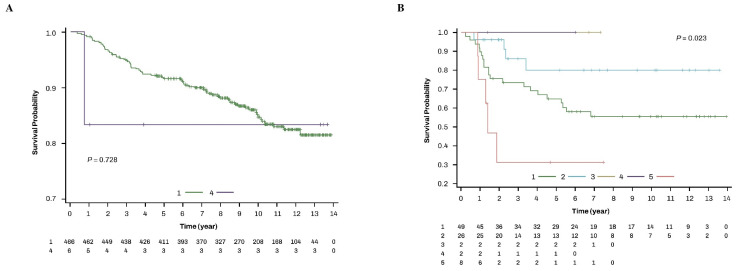
OS of adjuvant radiotherapy and chemoradiotherapy according to the type of histology in women with high-risk EEC. (**A**) endometrioid type and (**B**) non-endometrioid types. 1. adjuvant EBRT ± VB; 2. adjuvant VB alone; 3. adjuvant CCRT; 4. adjuvant chemotherapy after radiotherapy; 5. adjuvant radiotherapy after chemotherapy. CCRT, concurrent chemoradiotherapy; EBRT, external beam radiation therapy; EEC, early-stage endometrial cancer.

**Table 1 cancers-17-00187-t001:** Baseline characteristics in women with high-risk EEC.

Characteristics	No. (%)
No. of women	1341 (100%)
Age, mean (SD), y	58.2 ± 10.3
SES	
Mid- or high-SES	1191 (88.8)
Low SES	150 (11.2)
CCI	
0	883 (65.9)
1	132 (9.8)
2	245 (18.3)
3	56 (4.2)
≥4	25 (1.9)
Year of cancer diagnosis	
2007	88 (6.6)
2008	93 (6.9)
2009	94 (7.0)
2010	70 (5.2)
2011	95 (7.1)
2012	86 (6.4)
2013	83 (6.2)
2014	82 (6.1)
2015	75 (5.6)
2016	123 (9.2)
2017	115 (8.6)
2018	150 (11.2)
2019	187 (14.0)
Histology	
Endometrioid	794 (59.2)
Non-endometrioid	
Serous	281 (21.0)
Clear cell	35 (2.6)
Carcinosarcoma	201 (15.0)
Undifferentiated/dedifferentiated	30 (2.2)
Types of treatments	
Primary surgery alone	299 (22.3)
Primary surgery + adjuvant radiotherapy	541 (40.3)
Primary surgery + adjuvant chemotherapy	392 (29.2)
Primary surgery + adjuvant chemoradiotherapy	18 (1.3)
Primary surgery + adjuvant hormone therapy	91 (6.8)
Primary surgery	
Total hysterectomy	623 (46.5)
Radical hysterectomy	718 (53.5)
Adjuvant radiotherapy	
EBRT ± VB	515 (38.4)
VB alone	26 (1.9)
Adjuvant chemotherapy	
Platinum	341 (25.4)
Other agents	51 (3.8)
Adjuvant chemoradiotherapy	
CCRT	2 (0.2)
EBRT ± VB	1 (0.1)
VB alone	1 (0.1)
Sequential therapy	
Chemotherapy after radiotherapy	8 (0.6)
EBRT ± VB	6 (0.5)
VB alone	2 (0.2)
Radiotherapy after chemotherapy	8 (0.6)
EBRT ± VB	5 (0.4)
VB alone	3 (0.2)
Adjuvant hormone therapy	
Progestins	78 (5.8)
Tamoxifen	8 (0.6)
Progestins and Tamoxifen	0 (0)
Aromatase inhibitors	3 (0.2)
Gonadotropin-releasing hormone agonists	2 (0.2)
Postoperative VTE	
(−)	1290 (96.2)
(+)	51 (3.8)
Prophylactic anticoagulants	
(−)	862 (64.3)
(+)	479 (35.7)

CCI, Charlson comorbidity index; CCRT, concurrent chemoradiotherapy; EBRT, external beam radiation therapy; EEC, early-stage endometrial cancer; No, number; SD, standard deviation; SES, socioeconomic status; VB, vaginal brachytherapy; VTE, venous thromboembolism.

**Table 2 cancers-17-00187-t002:** Associations between the types of treatments and OS in women with high-risk EEC.

Variable	Univariable Analysis		Multivariable Analysis	
	HR (95% CI)	*p* Value	HR (95% CI)	*p* Value
Age, y	1.064 (1.048–1.081)	<0.001	1.059 (1.041–1.077)	<0.001
SES				
Low SES	ref	0.743		
Mid- or high-SES	0.93 (0.601–1.437)			
CCI				
0	ref			
1	1.152 (0.726–1.827)	0.549		
2	1.087 (0.747–1.58)	0.663		
3	0.776 (0.342–1.76)	0.544		
≥4	1.215 (0.449–3.288)	0.702		
Types of treatments				
Primary surgery alone	ref		ref	
Primary surgery + adjuvant EBRT ± VB	0.569 (0.405–0.799)	0.001	0.794 (0.557–1.132)	0.203
Primary surgery + adjuvant VB alone	0.703 (0.255–1.938)	0.495	0.704 (0.254–1.949)	0.500
Primary surgery + adjuvant chemotherapy	0.63 (0.408–0.972)	0.037	0.772 (0.496–1.202)	0.252
Primary surgery + adjuvant chemoradiotherapy	1.889 (0.814–4.386)	0.139	3.083 (1.311–7.247)	0.010
Primary surgery + adjuvant hormone therapy	0.18 (0.065–0.497)	0.001	0.464 (0.164–1.313)	0.148
Postoperative VTE				
(−)	ref	<0.001	ref	<0.001
(+)	3.537 (2.269–5.515)		2.75 (1.675–4.513)	
Prophylactic anticoagulants				
(−)	ref	<0.001	ref	0.222
(+)	1.774 (1.327–2.372)		1.228 (0.883–1.707)	

CCI, Charlson comorbidity index; CI, confidence interval; EBRT, external beam radiation therapy; EEC, early-stage endometrial cancer; HR, hazard ratio; SES, socioeconomic status; VB, vaginal brachytherapy; VTE, venous thromboembolism.

**Table 3 cancers-17-00187-t003:** Associations between adjuvant radiotherapy and chemoradiotherapy and OS in women with high-risk EEC.

Variable	Univariable Analysis		Multivariable Analysis	
	HR (95% CI)	*p* Value	HR (95% CI)	*p* Value
Age, y	1.064 (1.048–1.081)	<0.001	1.068 (1.047–1.099)	<0.001
SES				
Low SES	ref	0.743		
Mid- or high-SES	0.93 (0.601–1.437)			
CCI				
0	ref			
1	1.152 (0.726–1.827)	0.549		
2	1.087 (0.747–1.58)	0.663		
3	0.776 (0.342–1.76)	0.544		
≥4	1.215 (0.449–3.288)	0.702		
Adjuvant radiotherapy and chemoradiotherapy				
EBRT ± VB	ref		ref	
VB alone	1.254 (0.46–3.417)	0.659	0.828 (0.298–2.3)	0.717
CCRT	0	0.984	0	0.984
Chemotherapy after radiotherapy	0.999 (0.139–7.176)	0.999	1.534 (0.212–11.104)	0.672
Radiotherapy after chemotherapy	10.721 (4.282–26.84)	<0.001	11.87 (4.595–30.664)	<0.001
Postoperative VTE				
(−)	ref	<0.001	ref	0.001
(+)	3.537 (2.269–5.515)		3.441 (1.65–7.176)	
Prophylactic anticoagulants				
(−)	ref	0.001	ref	0.868
(+)	1.774 (1.327–2.372)		1.045 (0.623–1.751)	

CCI, Charlson comorbidity index; CCRT, concurrent chemoradiotherapy; CI, confidence interval; EBRT, external beam radiation therapy; EEC, early-stage endometrial cancer; HR, hazard ratio; SES, socioeconomic status; VB, vaginal brachytherapy; VTE, venous thromboembolism.

## Data Availability

All data generated or analyzed during this study are included in this published article and its information files. The data that support the findings of this study are available from the Health Insurance Review and Assessment Service (HIRA), but restrictions apply to the availability of these data, which were used under license for the current study and so are not publicly available. Data are, however, available from the authors upon reasonable request and with the permission of the HIRA.

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
