# Peer review of "Impacts of Postoperative Adjuvant Therapies on the Survival of Women with High-Risk Early-Stage Endometrial Cancer: A Cohort Study"

_cancers, 2025, doi:10.3390/cancers17020187_

Round 1
Reviewer 1 Report
Comments and Suggestions for Authors
The manuscript "Impacts of Postoperative Adjuvant Therapies on the Survival of Women with High-Risk Early-Stage Endometrial Cancer: A Cohort Study" represents an original article. Authors have evaluated the overall survival in 1341 women who were diagnosed with high-risk endometrial cancer at early stage and underwent surgery in combination with different adjuvant therapies, including adjuvant external beam radiation therapy, vaginal brachytherapy, combined radiation treatment, adjuvant chemotherapy, adjuvant hormone therapy and adjuvant chemoradiotherapy.
As endometrial cancer is the most common gynecological malignancy and early diagnostics is a desired goal in oncology, the topic undoubtedly is timely and important. The content of the manuscript corresponds to the scope of the journal “Cancers” and the section “Cancer Therapy”. The manuscript is detailed, rich in data and written in good English. Considering the relatively high incidence and prevalence of endometrial cancer, as well as the wide scope of postoperative adjuvant therapies for this tumour, the reported data could be of interest for the global scientific community.
Few corrections could be recommended, please:
1) In the lines 97 – 99, authors have stated “Women with high-risk EEC were defined as women with non-endometrioid histologic types, adjuvant EBRT ± VB, adjuvant chemotherapy, adjuvant chemoradiotherapy, or adjuvant hormone therapy”. Thus, only non-endometrioid histology is noted here as the histological inclusion criterion. However, in Table 1, there are 794 patients with high-risk early stage endometrial carcinoma. Patients having endometrioid histology are described also later in line 153; Tables S3 and S4; lines 171 – 178; Figure 2 and elsewhere in the text. Please, explain this controversy in regard to inclusion of endometrioid carcinoma.
2) Explain, please, if Stage I patients with G3 endometrioid histology, associated with myometrial invasion exceeding 50% or lymphovascular invasion were included in the high-risk early stage group in the current study. If G3 endometrioid histology was unrecognizable by the current study design, please, discuss, how this could influence your findings and conclusions.
3) Add, please, a concise and clear pathophysiological explanation on the few regimens associated with adverse prognosis: why radiotherapy after chemotherapy and adjuvant chemoradiotherapy might lead to/be associated with worse overall survival.
Finally, I would like to thank the authors for their contribution. It was a true pleasure to review this manuscript.
Reviewer 2 Report
Comments and Suggestions for Authors
In this work, the authors reported “Impacts of Postoperative Adjuvant Therapies on the Survival of Women with High-Risk Early-Stage Endometrial Cancer: A Cohort Study”. The study was analyzed appropriately and written in a good manner. However, some minor corrections should be addressed before publication.
Comments
1. Authors can be considered to write “simple summary” into “summary”. Furthermore, the first two lines of the summary and abstract need to be revised. Moreover, do not write the content with the same sentence in both places.
2. Line no.16, what was the impact of adjuvant radiotherapy after chemotherapy?
3. Line no.17, the “OS” should be explained in the first use.
4. Introduction: Authors are suggested to recheck the content, owing to the treatment strategies somewhere the meaning was missing. For example, Line no. 52-55. It was confusing.
5. Figures 2 and 3. Please draw them with high resolution.
6. Line no. 346-349, Incomplete
7. Line no. 349-350, “Nevertheless, large-scale studies are warranted to confirm these results.” What do the authors want to convey the message to the audience?
8. The research question raised in the introduction should be answered in the conclusions.
